# Enhancing farmed striped catfish (*Pangasianodon hypophthalmus*) robustness through dietary β-glucan

Sheeza Bano[1], Noor Khan[1], Mahroze Fatima[1], Anjum Khalique[2], Murat Arslan[3], Sadia Nazir[1], Muhammad Asghar[1], Ayesha Khizar[1], Simon John Davies[4], Alex H. L. Wan[4]*

1 Department of Fisheries & Aquaculture, University of Veterinary and Animal Sciences, Lahore, Pakistan,
2 Department of Animal Nutrition, University of Veterinary and Animal Sciences, Lahore, Pakistan,
3 Department of Aquaculture, Faculty of Fisheries, Ataturk University, Erzurum, Turkey, 4 Aquaculture and Nutrition Research Unit (ANRU), Carna Research Station, Ryan Institute and School of Natural Sciences, University of Galway, Carna, Connemara, Co. Galway, Ireland

* alex.wan@universityofgalway.ie

**Data Availability Statement:** All relevant data are within the manuscript and its Supporting information files.

## Abstract

β-glucan is a well-documented feed additive for its potent immunostimulatory properties in many farmed fish species. This study examined how it can also be a promising growth promoter, modulate antioxidant enzyme activities, and act as an anti-stress agent in striped catfish (*Pangasianodon hypophthalmus*). A 12-week feeding experiment was untaken to determine the effects of dietary β-glucan supplementation at graded levels (0, 0.5, 1.0, and 1.5 g kg$^{-1}$). Measured indicators suggest that a dietary inclusion level of 1.5 g kg$^{-1}$ β-glucan gave the highest positive responses: weight gain (120.10 g fish$^{-1}$), survival (98.30%), and lower FCR (1.70) ($P$<0.05). Whole body proximate analysis had only revealed that crude protein was significantly affected by the dietary inclusion of β-glucan ($P$<0.05), with the highest protein content (19.70%) being in fish that were fed with 1.5 g kg$^{-1}$ β-glucan. Although other inclusion levels (i.e., 0.5 and 1 g kg$^{-1}$) of β-glucan did not enhance body protein content ($P$>0.05). The assessment of fatty acid composition in muscle, liver, and adipose tissues showed modifications with the inclusion of β-glucan. Antioxidative-related enzyme activities (inc. catalase, glutathione peroxidase, and superoxide dismutase) that were measured in the liver had higher levels when fed with β-glucan inclusion diets ($P$<0.05). Following the feed trial, fish were subjected to crowding stress treatment. It was subsequently found that catfish fed with β-glucan-based diet groups had lower levels of blood stress-related indicators compared to the control group with no dietary β-glucan. The use of 1.5 g kg$^{-1}$ of dietary β-glucan resulted in the lowest measured levels of cortisol (43.13 ng mL$^{-1}$) and glucose (50.16 mg dL$^{-1}$). This study has demonstrated that the dietary inclusion of β-glucan can have functional benefits beyond the immunological enhancements in striped catfish. Furthermore, its use can increase production levels and mitigate the stress associated with intensive farming practices.

**Funding:** This study was funded by Punjab Agricultural Research Board (PARB), Pakistan-Project: Interactive Effects of Manipulated Artificial Feeds on Growth and Breeding Potential of Channa spp. The funders had no role in study design, data collection and analysis, decision to publish, or preparation of the manuscript. Open access was funded by IREL.

**Competing interests:** The authors declare there are no competing interests.

## 1. Introduction

As the human population grows, there is a greater need for food, but there is also a need for the food to be sustainably produced, limiting its impact on the environment and natural resources. Given their high feed conversion efficiencies and tolerance to intensive farming practices (e.g., higher stock densities) can provide a sustainable proteinous food source. However, intensive fish farming practices can induce a number of stressors on the animals during the production cycle, e.g., handling, hierarchy, and environmental stressors. These stressors can directly or indirectly affect fish health and compromise its ability to tolerate further stress [1]. Glucans (β-glucan) are a group of high molecular weight glucose polymers that are commercially used as an immune stimulatory feed additive [2–4]. These highly functional polysaccharides can be derived from yeast, fungi, algae, bacteria, and cereals. The immunological stimulatory bioactivity relates to the chemical similarities to fungal and bacteria cell wall signally molecules that are recognised by the immune system. The recognition of β-glucan by cell receptors in the gastrointestinal tract stimulates an inflammatory cascade response, increasing antibody production and resistance to pathogens [5, 6]. Other functional benefits to farmed fish have also been shown in β-glucan. For example, the compound can increase the activity of antioxidative-related enzymes (e.g., catalase, glutathione peroxidase, and superoxide dismutase) and subsequently lead to a reduction of measured reactive oxygen species (ROS) [7].

Past studies have shown that dietary use of β-glucan can enhance disease resistance in commercially important farmed species, such as common carp (*Cyprinus carpio* [8]), rainbow trout (*Oncorhynchus mykiss* [9]), Atlantic salmon (*Salmo salar* [10]), and sea bream (*Sparus aurata*, [11] and are employed in commercial aquafeeds. The response from the fish to dietary β-glucan can be variable depending on the farmed species, age class, and the duration of the compound being used. Depending on the β-glucan source, the molecular structure and binding abilities to the cell receptors in the fish. This would bring about different physiological responses in the animal [12]. Furthermore, these differences can result in variability in the optimal inclusion level to achieve the highest desired response [13].

Wild striped catfish (*Pangasianodon hypophthalmus*) are found in tropical freshwater environments in the Mekong River, Vietnam and Chao Phraya River, Thailand [14]. It is also a commercially farmed species, and in 2021, over 2.5 million tonnes of striped catfish were estimated to have been produced [15]. Due to its high growth rate, ease of rearing, and ability to take in oxygen from the air, the farming of this species has spread to include nations such as those in the Middle East. It is now a growing farmed fish species with the potential to provide a sustainable food source for the region. However, the need for sustainable food sources can comprise animal health and productivity rates. The present study seeks to understand how β-glucan can benefit farmed striped catfish other than its well-documented immunostimulation effect on the fish. The study will examine how β-glucan can influence fish growth performance, proximate composition, fatty acid profile, enzyme activities and relating to the antioxidative response. A post-feed trial study was also undertaken to understand further whether β-glucan can benefit farmed striped catfish after crowding stress.

## 2. Materials and methods

### 2.1 Test feed design and production

The design of the feed trial was to establish the optimum level of β-glucan. A previous study by Bano et al. [16] showed that 1 g kg$^{-1}$ elicited a positive response. Similarly, other studies

found the best level was 1 g kg-1 compared to other lowest or highest levels [17–21]. Therefore, the following graded levels were tested: 0 (G0), 0.5 (G0.5), 1.0 (G1.0), and 1.5 (G1.5) g kg$^{-1}$ using β-glucan derived from yeast (Food Chem, Huzhou, China). Test diets were prepared from a basal diet formulation (30% crude protein) with the graded levels of β-glucan. The basal diet formulation and proximate composition of the basal diet and fatty acid profiles are presented in Tables 1 and 2, respectively. All feed ingredients were ground to a particle size of less than 1.0 mm using an electric mixer. Feed ingredients were mixed thoroughly, and feeds were pelletised using a meat extruder (ANEX Model AG3060, Kowloon, Hong Kong). Feed pellets (3.0 mm diameter) were air-dried, packed, and stored at -20˚C for later use and analysis.

## 2.2 Fish and trial facility

Striped catfish (~250 g fish$^{-1}$ initial weight; ~seven months; fed with a 30% crude protein commercial diet) was sourced from the Department of Fisheries and Aquaculture, University of Veterinary and Animal Sciences (Ravi Campus, Pattoki, Pakistan). Fish were kept in concrete tanks for two weeks before the commencement of the feed trial. Fish were then

**Table 1. Diet formulation and proximate composition of the basal diet (%, dry weight).**

| *Diet formulation* | |
| --- | --- |
| Fish meal[a] | 30.00 |
| Soybean meal[a] | 16.00 |
| Corn gluten[a] | 11.00 |
| Wheat flour[b] | 20.00 |
| Rice polish | 15.00 |
| Sunflower oil | 6.00 |
| Vitamin premix[c] | 1.00 |
| Mineral premix[d] | 1.00 |
| *Nutritional composition* | |
| Moisture | 8.00 |
| Protein | 29.72 |
| Lipid | 9.92 |
| Ash | 10.00 |
| Fibre | 3.10 |
| Energy, MJ kg$^{-1}$ | 19.52 |
| Protein/Energy ratio, mg kJ$^{-1}$ | 16.01 |

[a] Fish meal (50% CP), soybean meal (45% CP), corn gluten (60% CP), and rice polish Purchased from Aqua Feeds Pvt Ltd Muktan, Pakistan.

[b] Family Flour Mill, Pattoki, Pakistan.

[c] Vitamin mixture: Vitamin A 3,500,000 IU kg$^{-1}$, vitamin $B_1$ 3,500 mg kg$^{-1}$, Vitamin $D_3$ 1,750,000 IU kg$^{-1}$, Zn gluconate 40 g kg$^{-1}$, vitamin E 3.500 mg kg$^{-1}$, vitamin PP (nicotinamide) 30 g kg$^{-1}$, sorbitol 20 g kg$^{-1}$ (Fivevet, Central Veterinary Medicine JSC No. 5, Ha Noi, Vietnam).

[d] Mineral mixture: Ferrous sulphate 25 g kg$^{-1}$, calcium phosphate 397 g kg$^{-1}$, calcium lactate 327 g kg$^{-1}$, magnesium sulphate 137 g kg$^{-1}$, sodium chloride 60 g kg$^{-1}$, potassium chloride 50 g kg$^{-1}$, potassium iodide 150 mg kg$^{-1}$, manganese oxide 800 mg kg$^{-1}$, copper sulphate 780 mg kg$^{-1}$, zinc oxide 1.5 g kg$^{-1}$, cobalt carbonate 100 mg kg$^{-1}$, manganese oxide 800 mg kg$^{-1}$, sodium selenite 20 mg kg$^{-1}$ (Fivevet, Central Veterinary Medicine JSC No. 5, Ha Noi, Vietnam).

**Table 2. Fatty acid composition of the basal diet (% of the total fatty acids).**

| | |
|---|---|
| 14:0 | 0.80 |
| 16:0 | 6.30 |
| 16:1n-7 | 0.10 |
| 18:0 | 3.10 |
| 18:1n-9 | 32.80 |
| 18:2n-6 | 55.50 |
| 18:3n-3 | 0.10 |
| 20:0 | <0.05 |
| 20:2n-6 | <0.05 |
| 20:4n-6 | <0.05 |
| 20:5n-3 | 0.10 |
| 22:6n-3 | 1.20 |
| ΣSFA | 10.20 |
| ΣMUFA | 32.80 |
| ΣPUFA | 56.90 |
| Σn3 | 1.40 |
| Σn6 | 55.60 |
| n3/n6 | 0.03 |

transferred and acclimated into an earthen pond (half an acre with 1.2 m depth). Groups of fifteen fish (250.37 ± 0.82 g fish$^{-1}$) were randomly allocated into twelve cages [hapas, 245 (L) ×180 (W) × 90 (D) cm] and placed into the pond. The hapas were made of synthetic green nylon net and designed to have two nested compartments tied to a bamboo pole frame. The inner compartment had a mesh size of 1.1 cm that held the fish. Uneaten pellets pass to the outer compartment, composed of a finer mesh netting (1.5 mm). The top of the cages was covered with another net to prevent escape and predation.

Allocation of each test feed was randomly assigned to each cage to give a triplicate treatment design (n = 3). Water exchanges were performed daily on the pond to prevent excess build-up of waste nutrients using well water. Paddle wheels were used to maintain aeration in the pond system cages. The water quality parameters were measured at 08:00 am, before feeding, including the water temperature (30 ±0.31˚C, standard deviation, SD), pH (6.32 ±0.17, SD), dissolved oxygen (5.60 ±0.25 mg L$^{-1}$, SD) and total dissolved solids (1320 ±0.11 mg L$^{-1}$, SD) using the Hanna HI 9828/4-01 multi-parameter (Chelmsford, UK). The fish were fed twice daily (09:00 and 16:00) at 3% of the body weight. After twenty minutes of giving the feed ration, the uneaten feed was collected from the outer compartment of hapas to ascertain net feed consumption. Fish were weighed fortnightly, and the feed rations to each cage were adjusted to reflect the biomass increase.

### 2.3 Animal ethics

The research protocols and procedures used on this fish trial was approved by the animal use and animal care committee of the University of Veterinary and Animal Sciences, Lahore, Pakistan (DR/161, 26-04-2021).

## 2.4 Growth performance, survival, and feed utilisation

After the twelve-week feed trial, feed rations were withheld for 24 hr to allow gut clearance, counted, and weighed for the following calculations.

$$\text{Weight gain, g fish}^{-1} = \text{final weight (g)} - \text{initial weight (g)} \tag{1}$$

$$\text{Feed conversion ratio (FCR)} = \text{feed consumed (g)}/\text{wet weight gained (g)} \tag{2}$$

$$\begin{aligned}\text{Specific growth rate (SGR), \% day}^{-1} \\ = 100 \times (\ln(\text{final weight (g)}) - \ln(\text{initial weight (g)}))/\text{days}\end{aligned} \tag{3}$$

$$\text{Survival, \%} = 100 \times (\text{number of survived fish}/\text{initial number of fish}) \tag{4}$$

## 2.5 Proximate composition

Samples of formulated feed and experimental fish were examined for moisture, crude protein, crude fat, and ash by following the protocol of the Association of Official Analytical Chemists [22]. For sample collection, two fish from each cage were randomly captured and euthanised with clove oil (0.40 mL L$^{-1}$) and stored samples at -20˚C until the analysis. For the moisture content, whole-body fish samples were oven-dried (Universal Oven UN260, Memmert, Germany) at 105˚C until a constant weight was obtained. The protein content (N × 6.25) was ascertained by the Kjeldahl apparatus (Kjeltec 8100, FOSS, Hilleroed, Denmark). Crude fat was extracted by using the Soxhlet apparatus (R106S, Behr Labor-Technik, Dusseldorf, Germany), and ash contents were determined by using a muffle furnace at 550˚C (TMF-3100, Eyela Co., Tokyo, Japan).

## 2.6 Fatty acid profile

At the end of the trial, two additional fish from each cage were randomly sampled and euthanised with clove oil (0.40 mL L$^{-1}$). The fish were then dissected to obtain the muscle, liver, and adipose tissues. Samples were grounded and stored at -80˚C until it was later used for fatty acids analysis. Lipid extraction of test diets and fish samples was performed according to the procedure of [23] by homogenising the samples in a solution of chloroform:methanol (2:1, v/v). Samples were then esterified into fatty acid methyl esters (FAME) [24], and subsequently analysed on a gas chromatography flame ionisation detector (Agilent 6890N, Santa Clara, USA). Fatty acids were identified by comparing the known fatty acids standards (Supelco 37 Component FAME Mix, Sigma-Aldrich, Darmstadt, Germany).

## 2.7 Antioxidant enzyme activity

At the end of the experiment, two additional fish were randomly sampled from each cage and euthanised using clove oil (0.40 mL L$^{-1}$). The euthanised fish were sampled for the liver, and muscle tissues were removed and kept at -80˚C until later analysis on antioxidant enzyme activity. The tissue samples (1/4, w/v) were homogenised in nine volumes of 20 mM phosphate buffer (pH, 7.4), 0.1% Triton X-100, and 1 mM EDTA. The homogenates were centrifuged for 10 minutes (860 $g$ at 4 ºC) to remove the solid particulates (5810R, Eppendorf centrifuge, Hamburg, Germany) and the supernatant was removed and stored at -80 ºC for later analysis [25]. Catalase enzyme activity (CAT; EC 1.11.1.6) activity was accessed by following the protocol [26]. Glutathione peroxidase activity (GPx; EC 1.11.1.9) was evaluated according to Flohé

and Günzler methodology [27]. Superoxide dismutase activity (SOD; EC 1.15.1.1) was measured following the method described by Giannopolitis and Ries [28].

## 2.8 Crowding stress assessment

After the twelve-week feeding trial, blood samples were collected from two fish per cage as a baseline indicator before the fish were subjected to a crowding stress assessment. The remaining fish from each rearing cage was then transferred to 12 test tanks (6 or 7 survival) corresponding to the dietary treatments. For stress induction, fish were exposed to stress for 2 hr by reducing 90% of the water level at a depth of 50 cm [29]. After 2 hr of stress, the blood was collected from 2 fish per tank (6 per treatment) at the end of this period. Then, the water was returned to the normal level and continuously aerated to maintain the dissolved oxygen at 5.8 mg L$^{-1}$. Blood samples from two fish in each tank were collected randomly and sampled after 8 hr and 24 h of stress induction. Fish sampled after each blood sampling (2, 8, and 24 hr) were kept in a separate tank and a new fish was sampled subsequently. Fish were anaesthetised with clove oil (0.20 ml L$^{-1}$) before blood samples were taken at the caudal ventral vein 5 min after capture. Blood samples were kept at room temperature for 1 hr to allow clotting and centrifuged at 3,000 $g$ at 4 ℃ for 15 min. Separated serum was removed for storage and kept at -80 ℃ for later analysis on cortisol and glucose levels as indicators to stress [30]. For serum cortisol, an enzyme-linked immunosorbent assay (ELISA) kit was used to determine cortisol levels (DKO00, Diametra, Milano, Italy [31]). Measurements of the serum glucose levels were based on the biuret method and were performed through a commercial assay kit (Pars Azmun, Tehran, Iran).

## 2.9 Statistical analysis

The results were presented as mean values with the corresponding standard deviation. Datasets were assessed for statistical significance using one-way analysis of variance (ANOVA). Posthoc Tukey test was undertaken to identify statistical differences between dietary treatments (SAS 9.1 Software package, Cary, North Carolina, U.S.A). Two-way ANOVA was used to determine the statistical significance of diet, time, and diet x time on stress indicators- glucose and cortisol, and posthoc Tukey test was conducted to discern treatment x time effect. A principal component analysis was undertaken to visualise and discern possible relationships in the measured fatty acids. Differences were considered statistically significant when was $P<0.05$ [32].

# 3. Results

## 3.1 Growth performance, feed utilisation, and survival

At the end of the feeding trial, growth performance, feed conversion ratio (FCR), and survival were significantly influenced by the dietary treatments ($P<0.05$). There was a trend where growth performance indicators were enhanced as the dietary β-glucan dosage increased. The highest mean final weight, weight gain, and specific growth rate were observed in fish fed with 1.5 g kg$^{-1}$ β-glucan diet. While fish fed with the dietary control had the lowest growth measurements, e.g., weight gain (90 g), specific growth rate (3.74%), and feed conversion ratio (2.11). Fish fed with a β-glucan inclusion level of 1.5 g kg$^{-1}$ had the lowest FCR, and the control group had the highest values, which were 1.70 and 2.11, respectively. Fish fed with dietary β-glucan supplementation, regardless of the inclusion levels, had significantly higher growth and lower FCR in comparison to the control fish (P<0.022). Although the survival rate was not influenced by the dietary treatments and on average, the survival was 96.1% (Table 3).

**Table 3. Growth performance and feed utilisation of striped catfish (*Pangasianodon hypophthalmus*) fed with graded levels of β-glucan for 12 weeks (n = 3).**

| | Test diets | | | | |
|---|---|---|---|---|---|
| | G0 | G0.5 | G1.0 | G1.5 | P-value |
| Mean initial weight; g fish[-1] | 250.20±0.50 | 250.26±1.00 | 250.50±1.50 | 250.50±0.50 | *0.860* |
| Mean final weight; g fish[-1] | 340.50±5.70[c] | 354.60±2.80[b] | 363.00±2.00[ab] | 370.30±5.90[a] | *<0.001* |
| Mean weight gain; g fish[-1] | 90.00±6.20[c] | 104.33±1.80[b] | 112.50±3.50[ab] | 120.10±6.10[a] | *<0.001* |
| SGR; % day[-1] | 3.74±0.05[c] | 3.86±0.01[b] | 3.92±0.07[b] | 4.01±0.04[a] | *0.020* |
| FCR | 2.11±0.05[a] | 1.90±0.03[b] | 1.81±0.04[c] | 1.70±0.02[d] | *0.022* |
| Survival; % | 93.30±0.00 | 95.60±3.80 | 97.80±3.80 | 97.80±3.80 | *0.100* |

Data are presented as mean values with ± SD. Different superscripts on the same row indicate there is a significant difference ($P<0.05$, ANOVA). SGR: specific growth rate; FCR: feed conversion ratio. Dietary treatments: 0, (G0); 0.5 (G0.5); 1.0 (G1.0); 1.5 (G1.5) g kg[-1] β-glucan inclusion level.

## 3.2 Proximate composition

With the exception of the crude protein content ($P<0.05$), dietary treatments did not significantly influence the proximate body composition in striped catfish (Table 4, $P>0.05$). Fish fed with a 1.5 g kg[-1] β-glucan diet had the highest crude protein content (19.70%), while those fed the control diet had the lowest value (17.60%).

## 3.3 Fatty acid profile

The test diets did not significantly influence the fatty acid profile in either the muscle, liver, and adipose tissues (Tables 5–7). Saturated fatty acids (SFA) were highest in the liver, while monounsaturated fatty acids (MUFA) and polyunsaturated fatty acids (PUFA) were dominant in adipose and muscle, respectively. The n3/n6 ratio was 0.05, 0.10, and <0.01 in muscle, liver, and adipose tissues. Linoleic acid (LA; 18:2n-6) was the most abundant fatty acid in muscle and liver. The fatty acid profile of adipose tissue was dominated by linoleic acid and oleic acid (OA; 18:1n-9). Both eicosapentaenoic acid (EPA; 20:5n-3) and docosahexaenoic acid (DHA; 22:6n-3) were highest in the liver (0.90 and 2.90%, respectively) and lowest in adipose (< 0.05%). Muscle EPA and DHA levels were 0.30 and 1.30%, respectively. PCA on the fatty acid composition showed there was no discrete grouping between dietary groups amongst the muscle, liver, and adipose tissues (Fig 1).

## 3.4 Antioxidant enzymes activities

At the end of the feed trial, antioxidative-related enzyme activities in the liver were significantly influenced by dietary β-glucan supplementations ($P<0.05$; Fig 2). The highest and

**Table 4. Proximate composition of striped catfish (*Pangasianodon hypophthalmus*) fed with graded levels of β-glucan for 12 weeks (n = 3, % of wet weight).**

| | Test diets | | | | |
|---|---|---|---|---|---|
| | G0 | G0.5 | G1.0 | G1.5 | P-value |
| Moisture | 73.91±0.10 | 73.24±0.10 | 73.12±0.10 | 72.90±0.20 | *0.086* |
| Crude protein | 17.60±0.40[b] | 18.10±0.60[b] | 18.70±0.80[ab] | 19.70±0.40[a] | *0.014* |
| Crude fat | 6.20±0.60 | 6.40±0.50 | 5.90±0.10 | 5.08±0.60 | *0.500* |
| Ash | 1.30±0.10 | 1.30±0.00 | 1.20±0.00 | 1.10±0.00 | *0.076* |

Data are presented as mean values with ± SD. Different superscripts on the same row indicate there is a significant difference ($P<0.05$). Dietary treatments: 0, (G0); 0.5 (G0.5); 1.0 (G1.0); 1.5 (G1.5) g kg[-1] β-glucan inclusion level.

**Table 5. Fatty acid profile of muscle in striped catfish (*Pangasianodon hypophthalmus*) fed with graded levels of β-glucan for 12 weeks (%, n = 3).**

| | Test diets | | | |
|---|---|---|---|---|
| | **G0** | **G0.5** | **G1.0** | **G1.5** |
| 14:0 | 1.50±0.30 | 1.05±0.30 | 1.60±0.40 | 1.50±0.20 |
| 16:0 | 25.20±0.5 | 25.01±0.70 | 25.20±0.20 | 25.30±0.40 |
| 16:1n-7 | 2.30±0.10 | 2.20±0.30 | 2.30±0.20 | 2.20±0.30 |
| 18:0 | 3.30±0.20 | 3.40±0.40 | 3.30±0.50 | 3.50±0.40 |
| 18:1n-9 | 29.40±0.50 | 29.30±0.60 | 29.30±0.80 | 29.40±0.30 |
| 18:2n-6 | 33.50±0.40 | 33.80±0.70 | 33.50±0.50 | 33.40±0.60 |
| 18:3n-3 | 0.10±0.10 | 0.20±0.20 | 0.10±0.00 | 0.10±0.00 |
| 20:0 | 0.20±0.20 | 0.20±0.00 | 0.20±0.00 | 0.20±0.10 |
| 20:2n-6 | 1.20±0.20 | 1.10±0.20 | 1.20±0.10 | 1.20±0.20 |
| 20:4n-6 | 1.60±0.30 | 1.50±0.20 | 1.50±0.30 | 1.50±0.10 |
| 20:5n-3 | 0.30±0.20 | 0.30±0.10 | 0.30±0.10 | 0.30±0.00 |
| 22:6n-3 | 1.30±0.20 | 1.30±0.10 | 1.40±0.30 | 1.30±0.20 |
| ΣSFA | 30.20±0.40 | 30.20±0.50 | 30.30±0.50 | 30.60±0.40 |
| ΣMUFA | 31.70±0.50 | 31.50±0.10 | 31.60±0.60 | 31.60±0.40 |
| ΣPUFA | 38.10±0.20 | 38.30±0.30 | 38.10±0.60 | 37.80±0.70 |
| n3 | 1.80±0.10 | 1.80±0.20 | 1.80±0.10 | 1.80±0.10 |
| n6 | 36.30±0.30 | 36.50±0.20 | 36.20±0.20 | 36.00±0.40 |
| n3/n6 | 0.05±0.01 | 0.05±0.01 | 0.05±0.00 | 0.05±0.01 |

Data are presented as mean values with ± SD. Dietary treatments: 0, (G0); 0.5 (G0.5); 1.0 (G1.0); 1.5 (G1.5) g kg$^{-1}$ β-glucan inclusion level.

lowest catalase activities were observed in fish fed G1.5 (71.20 U mg$^{-1}$) and G0 (control) diets (54.10 U mg$^{-1}$), respectively. Fish fed β-glucan supplemented diets had significantly higher CAT activities than those fed on the control diet in all circumstances. Glutathione peroxidase activity was significantly higher in the G1.5 dietary fish group (61.30 U mg$^{-1}$) than in those fed the other three experimental diets. Fish fed with the control diet had lower GPx activity (45.00 U mg$^{-1}$) than those fed β-glucan supplemented diets, while G0.5 and G1.0 fed fish exhibited similar GPx activities. Superoxide dismutase activity was significantly lower in fish-fed with the control diet in comparison to those fed β-glucan supplemented diets, which had similar SOD activities.

Regarding muscle antioxidant enzymes, CAT and GPx activities were significantly influenced by the dietary treatments ($P<0.05$), while there were no significant changes observed in SOD activity (Fig 2). Fish fed with G1.0 and G1.5 diets had significantly higher CAT activities in comparison to G0 and G0.5 dietary groups. Similar to the liver results, muscle GPx activity was the highest in the G1.5 dietary fish group (37.40 U mg$^{-1}$) compared to the other three test diets. Fish fed with the dietary control had significantly lower GPx activity (27.90 U mg$^{-1}$) than those fed β-glucan supplemented diets, while G0.5 and G1.0 fed fish exhibited similar GPx activities.

## 3.5 Crowding stress response

Test diets had significantly affected both serum cortisol and glucose levels in both fish pre and post-crowding stress ($P<0.001$, Table 8). Prior to the crowding stress, the lowest serum cortisol and glucose levels were found in fish fed with the G1.5 diet, while fish in the G0 (control)

**Table 6. Fatty acid profile of liver in striped catfish (*Pangasianodon hypophthalmus*) fed with graded levels of β-glucan for 12 weeks (%, n = 3).**

| | Test diets | | | |
|---|---|---|---|---|
| | G0 | G0.5 | G1.0 | G1.5 |
| 14:0 | 3.40±0.30 | 3.50±0.20 | 3.20±0.40 | 3.50±0.30 |
| 16:0 | 26.40±0.40 | 26.30±0.30 | 26.60±0.50 | 26.40±0.30 |
| 16:1n-7 | 1.80±0.20 | 2.00±0.20 | 1.90±0.10 | 2.00±0.20 |
| 18:0 | 2.60±0.20 | 2.60±0.10 | 2.60±0.10 | 2.50±0.20 |
| 18:1n-9 | 25.20±0.50 | 25.00±0.60 | 24.90±0.40 | 25.00±0.50 |
| 18:2n-6 | 31.50±0.40 | 31.20±0.20 | 31.20±0.50 | 31.40±0.30 |
| 18:3n-3 | 0.80±0.20 | 0.90±0.30 | 0.90±0.30 | 0.80±0.20 |
| 20:0 | 0.40±0.20 | 0.40±0.10 | 0.30±0.10 | 0.30±0.20 |
| 20:2n-6 | 1.70±0.20 | 1.80±0.30 | 1.80±0.50 | 1.90±0.40 |
| 20:4n-6 | 2.60±0.20 | 2.60±0.20 | 2.50±0.30 | 2.50±0.20 |
| 20:5n-3 | 0.80±0.20 | 0.90±0.30 | 1.00±0.30 | 0.90±0.20 |
| 22:6n-3 | 2.90±0.20 | 2.80±0.20 | 2.90±0.10 | 2.90±0.20 |
| ΣSFA | 32.80±0.20 | 32.80±0.10 | 32.80±0.10 | 32.70±0.20 |
| ΣMUFA | 27.00±0.30 | 26.90±0.50 | 26.80±0.50 | 27.00±0.30 |
| ΣPUFA | 40.20±0.40 | 40.20±0.30 | 40.40±0.30 | 40.40±0.40 |
| n3 | 4.50±0.20 | 4.60±0.30 | 4.90±0.50 | 4.60±0.60 |
| n6 | 35.80±0.40 | 35.60±0.30 | 35.50±0.20 | 35.80±0.30 |
| n3/n6 | 0.10±0.00 | 0.10±0.00 | 0.10±0.00 | 0.10±0.00 |

Data are presented as mean values with ± SD. Dietary treatments: 0, (G0); 0.5 (G0.5); 1.0 (G1.0); 1.5 (G1.5) g kg$^{-1}$ β-glucan inclusion level.

group had the highest measured values. These differences are inverse dose dependence relationships, where increasing β-glucan inclusion leads to decreasing stress indicator levels.

The effect of time did reduce the measured stress indicators (P<0.001). After inducing crowding stress (2 hr), fish in the G1.5 diet group had a difference of 17.48% lower cortisol when compared to fish fed with the G0 diet, while glucose levels were 21.12% lower. These trends were repeated in both 8 hr and 24 hr measurements. There was a stepwise decrease in mean cortisol and glucose levels. Only the dietary β-glucan treatment groups had returned to their pre-stress levels after 8 hr post crowding stress. In contrast, the control group did not return to their pre-stress levels at 24 hr of crowding. Two-way ANOVA showed an interaction effect between diet and time in cortisol and glucose levels, P = 0.022 and P<0.001, respectively.

## 4. Discussion

In the present study, dietary β-glucan had significantly improved growth performance and feed utilisation indices, body crude protein content, and antioxidant enzyme activities in striped catfish. The effects were most pronounced at the dosage of 1.5 g kg$^{-1}$. In comparison, β-glucan supplementation also improved the growth rate of other farmed aquatic species, including the Pacific white shrimp (*Litopenaeus vannamei* [33]) and freshwater gangetic mytus catfish (*Mystus cavasius* [34]). In contrast, other studies reported that dietary β-glucan had no influence on growth performance in Nile tilapia (*Oreochromis niloticus*, [35]) and channel catfish (*Ictalurus punctatus*) [36]. The exact mechanism by which β-glucan promotes growth is not fully understood. However, there is some indication that somatic growth is increased due to the glucanase synthesis that degrades the β-glucans to produce energy [37].

**Table 7. Fatty acid profile of adipose in striped catfish (*Pangasianodon hypophthalmus*) fed with graded levels of β-glucan for 12-weeks (%, n = 3).**

|  | Test diets | | | |
|---|---|---|---|---|
|  | G0 | G0.5 | G1.0 | G1.5 |
| 14:0 | 2.40±0.20 | 2.50±0.30 | 2.30±0.20 | 2.40±0.20 |
| 16:0 | 16.50±0.30 | 16.50±0.30 | 16.40±0.40 | 16.30±0.20 |
| 16:1n-7 | 2.30±0.10 | 2.30±0.00 | 2.30±0.00 | 2.30±0.00 |
| 18:0 | 4.60±0.00 | 4.60±0.10 | 4.70±0.1 | 4.70±0.00 |
| 18:1n-9 | 34.80±0.20 | 34.80±0.10 | 34.90±0.20 | 34.90±0.20 |
| 18:2n-6 | 35.80±0.20 | 35.90±0.20 | 35.80±0.20 | 35.80±0.10 |
| 18:3n-3 | 0.10±0.00 | 0.10±0.00 | 0.10±0.00 | 0.10±0.00 |
| 20:0 | 3.30±0.20 | 3.30±0.10 | 3.30±0.10 | 3.30±0.00 |
| 20:2n-6 | 0.10±0.00 | 0.10±0.00 | 0.10±0.00 | 0.10±0.00 |
| 20:4n-6 | 0.10±0.00 | 0.10±0.00 | 0.10±0.00 | 0.10±0.00 |
| 20:5n-3 | <0.05 | <0.05 | <0.05 | <0.05 |
| 22:6n-3 | <0.05 | <0.05 | <0.05 | <0.05 |
| ΣSFA | 26.80±0.20 | 26.80±0.30 | 26.70±0.20 | 26.60±0.20 |
| ΣMUFA | 37.20±0.10 | 37.30±0.20 | 37.20±0.20 | 37.20±0.20 |
| ΣPUFA | 36.00±0.20 | 36.00±0.20 | 36.10±0.10 | 36.10±0.20 |
| n3 | 0.10±0.00 | 0.10±0.00 | 0.10±0.00 | 0.10±0.00 |
| n6 | 35.90±0.20 | 35.90±0.10 | 36.00±0.20 | 36.00±0.20 |
| n3/n6 | <0.01 | <0.01 | <0.01 | <0.01 |

Data are presented as mean values with ± SD. Dietary treatments: 0, (G0); 0.5 (G0.5); 1.0 (G1.0); 1.5 (G1.5) g kg$^{-1}$ β-glucan inclusion level.

While the increase in growth and survival may be due to improved digestive function resulting in better nutrient absorption to meet greater dietary needs [38]. The attributing factor may result from selective fermentation in the polysaccharide by probiotic bacteria in the gastrointestinal tract. This can subsequently lead to better gut health and greater nutrient bioavailability. In the present study, dietary treatments did not affect the survival rate. Likewise, dietary β-glucan also did not influence the survival rate in juvenile Persian sturgeon (*Acipenser persicus* [21]). Although β-glucan supplementation did increase the survival rate of pompano fish (*Trachinotus ovatus*) when fed at an inclusion level of 1.0 g kg$^{-1}$ [39].

The whole-body fish composition is directly related to the composition of their feed and its nutritional value [40]. The dietary β-glucan supplementation did not modify the proximate composition except for crude protein in the current study. The present study found an increasing trend in the body protein content with increasing dietary β-glucans inclusion. In comparison, by adding β-glucan into to red sea bream (*Pagrus major*) diet, the fish body content also increased [41]. β-glucan supplementation in Caspian kutum (*Rutilus frisii kutum*) diet, the authors reported an improved crude protein content. Although the moisture, lipid, and ash contents were unchanged [42].

Modifying the fish fillet fatty acid profile by incorporating feed supplements is important because of its potential health benefits to consumers [43, 44]. The present study is the first to investigate the dietary effects of β-glucan on the fatty acid composition in fish tissue. Although in a previous study, dietary β-glucan modified the proportions of saturated and unsaturated fatty acids, which improved the meat flavour and quality in finishing pigs [45]. Compared to the current study, β-glucan supplementation did not influence the fatty acid profile in the muscle, liver, or adipose tissues.

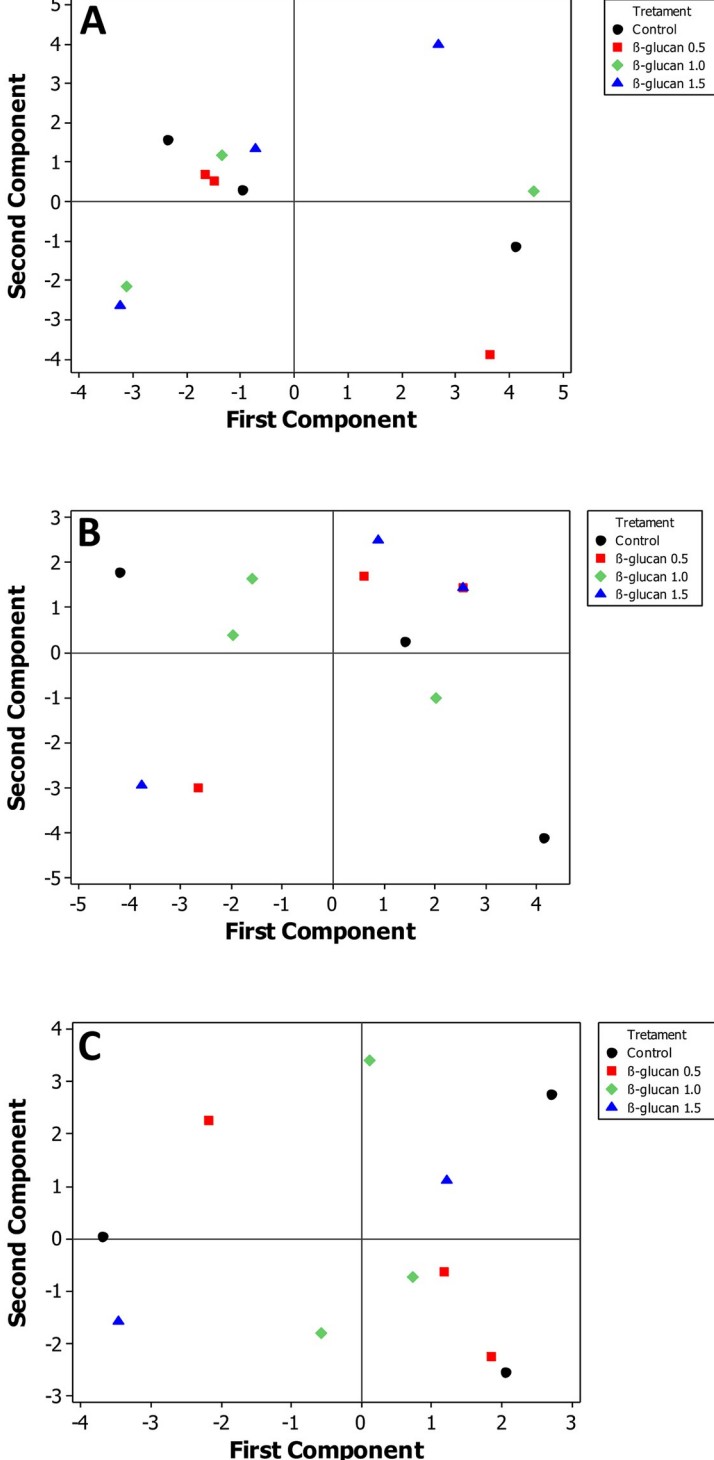

**Fig 1. Principal component analysis of the fatty acid composition in striped catfish (*Pangasianodon hypophthalmus*) after being fed with β-glucan supplemented diets- first and second component. A**) Muscle; **B**) Liver; **C**) Adipose tissue. Dietary treatment: ● 0, (G0); ■, 0.5, (G0.5); ◆ 1.0, (G1.0); ▲1.5, (G1.5) g kg$^{-1}$ glucan inclusion level. The first and second component explains 71.60, 69.60, and 51.80% of the cumulative error in the fatty acid composition of muscle, liver and adipose tissue, respectively.

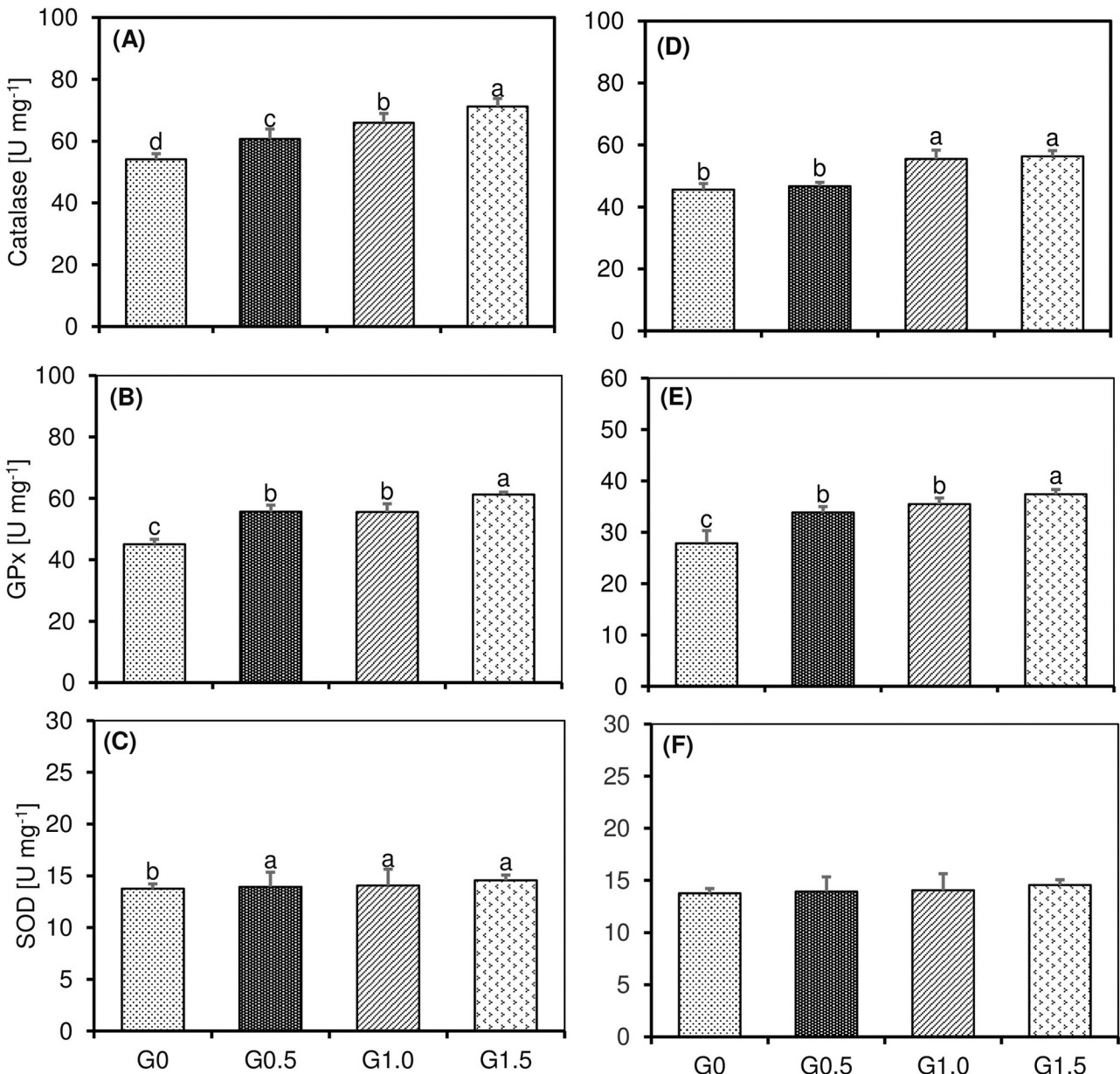

**Fig 2. Effect of β-glucan supplementation on antioxidant enzymes in muscle (A, B, and C) and liver (D, E, and F) of striped catfish (*Pangasianodon hypophthalmus*).** Results are presented as mean ±SD. Bars with different letters are significantly different (*P*<0.05). Dietary treatments: 0, (G0); 0.5 (G0.5); 1.0 (G1.0); 1.5 (G1.5) g kg⁻¹ β-glucan inclusion level.

It has been documented that there is a correlation between the stress the farmed fish experiences and its impact on its health status. Mitigating metabolic and physiological stress using functional feed ingredients, such as antioxidants, can modulate the fish's antioxidant enzymes and immune response [46]. While enhancing the antioxidative capacity without the need for antioxidants is also possible. This was observed in the current study, where both CAT, SOD, and GPx antioxidant enzymes measured in the fish liver and muscle were enhanced with increasing levels of dietary β-glucan. However, the dosages used in the present trial were insufficient to ascertain an optimum inclusion. This is due to no downward trend observed in the

**Table 8. Cortisol and glucose levels in striped catfish (*Pangasianodon hypophthalmus*) pre and post crowd stressing (n = 3).**

| hr | Diet | | | | P-value | | |
|---|---|---|---|---|---|---|---|
| | G0 | G0.5 | G1.0 | G1.5 | Diet | Time | Diet*Time |
| *Cortisol* [ng dL⁻¹] | | | | | | | |
| 0 | 55.73±2.45[d,e,f] | 48.8±2.02[f,g,h] | 45.46±2.19[g,h] | 43.20±2.48[h] | <0.001 | <0.001 | 0.022 |
| 2 | 75.5±2.92[a] | 69.86±2.01[a,b] | 63.73±1.27[b,c,d] | 62.33±1.45[b,c,d] | | | |
| 8 | 74.57±4.16[a] | 68.23±2.82[a,b,c] | 63.71±3.49[b,c,d] | 60.40±3.07[c,d,e] | | | |
| 24 | 67.86±3.37[a,b,c] | 52.42±2.43[e,f,g] | 46.63±4.74[g,h] | 43.13±1.20[h] | | | |
| *Glucose* [mg dL⁻¹] | | | | | | | |
| 0 | 61.16±2.02[f,g] | 59.33±1.15[g,h] | 53.67±1.53[i,j] | 52.47±1.48[i,j] | <0.001 | <0.001 | <0.001 |
| 2 | 73.50±1.40[a] | 68.20±1.62[b,c,d] | 74.3±2.29[a] | 71.35±2.11[ab] | | | |
| 8 | 67.43±2.00[b,c] | 67.26±2.52[c,d,e] | 59.50±1.17[g,h] | 58.4±2.23[g,h] | | | |
| 24 | 57.28±1.20[d,e,f] | 55.8±1.37[e,f] | 51.75±1.62[h,i] | 50.16±0.76[j] | | | |

Two-way ANOVA. Different superscripts represent the differences between groups and time ($P<0.05$).

higher β-glucan inclusion levels. Some past studies have shown an optimum inclusion level based on antioxidative-related assays, such as in red sea bream (*Pagrus major*), where the dietary level was at 0.5 g kg⁻¹ β-glucan [41]. Other studies have also observed that β-glucan addition can increase antioxidative levels, including CAT, SOD, and GPx, even after disease or stimulated disease-challenged [47–50]. Furthermore, even crude forms of β-glucan can also enhance antioxidative capacity. For example, using yeast cell walls partly composed of β-glucan can enhance the antioxidant enzyme activities [51, 52].

In the present study, the measured SOD activity was affected by dietary treatments in the muscle, where the activity level was significantly higher in β-glucan supplemented diets. Although there was no apparent change in the liver. Comparing this to past studies, this trend was found to be the opposite. For instance, the use of β-glucan had increased SOD activity levels in both Pacific red snapper (*Lutjanus peru*) [47] and gilthead seabream (*Sparus aurata*) [53].

Stress is a condition in which animal homeostasis is disturbed by intrinsic and extrinsic stimuli [54]. Serum cortisol and glucose levels are common parameters used as a measure of stress response in many animal studies [55]. Cortisol is the glucocorticoid where the circulating levels in the blood can increase dramatically under stressful conditions [56]. In the current study, the cortisol levels in the fish fed β-glucan supplemented diets were lower than the control, regardless of the sampling time interval. Crowding challenge increased the cortisol level in all treatments. The return to prestress levels was only observed in β-glucan dietary groups, while glucose levels also showed a similar trend. Past studies have shown a similar trend of β-glucan having the ability to lower cortisol in farmed fish species, including striped catfish that was fed with only 4 weeks of β-glucan [17, 57]. It was suggested that the use of β-glucan was suppressing some part of the immune function, though subsequent disease challenges did not find this to be the case [57]. Furthermore, in some species, the use of dietary β-glucan does not have an effect on cortisol levels, e.g., tilapia [58]. While in matrinxã (*Brycon amazonicus*), another freshwater species, dietary β-glucan had increased cortisol production and immune function [59]. This disparity and knowledge gap warrants further investigation, such as the systematic comparison of species through taxa and natural habitats using molecular biomarkers responsible for cortisol production and glucogensis.

It is important to note that the β-glucan used in the present study was based on the indicative levels from previous studies [16, 17, 20, 21]. These had focused on improving health

parameters rather than the current study's metrics. Therefore, there is a probability that the optimal/maximal level is not attained, which is the case in the present study. Testing higher dietary inclusion levels could identify this level. As such, it warranted on the basis of scientific completeness and addressing the practical question of whether the farmer would benefit from a higher production yield if they used a higher β-glucan level in the aquafeed. By identifying this maximal level, further data analysis could be of benefit to this question and draw an economic maximal level as well (i.e., the highest level to give the greatest economic benefit) including the establishment of a complete dose-response curve/model using linear regression and/or polynomial orthogonal contrast.

## 5. Conclusion

β-glucan is a potent and effective functional feed ingredient often used in aquafeeds as an elicitor for increasing immunological response. The present study assessed whether dietary β-glucan can offer additional benefits in farmed striped catfish. Using a dietary inclusion level of 1.5 g kg$^{-1}$ β-glucan in the feed produced the highest beneficial response in growth performance, feed utilisation, body protein accretion, and indicators of antioxidative function. More importantly, the greatest advantage of using dietary β-glucan in striped catfish is its ability to reduce the impact of husbandry stress indicators. However, the tested levels of β-glucan in the present study did not establish the maximal efficacy threshold. Therefore, there is an opportunity for further optimisation in striped catfish.

## Supporting information

**S1 File. Morphometric, fatty acid, proximate composition, stress indicator and liver antioxidative activity data.**
(XLSX)

## Acknowledgments

Special thanks to the Faculty of Fisheries, Ataturk University, Erzurum, Turkey, for the technical support in the fatty acids analysis.

## Author Contributions

**Conceptualization:** Sheeza Bano, Noor Khan, Mahroze Fatima.

**Data curation:** Sheeza Bano, Mahroze Fatima, Murat Arslan, Alex H. L. Wan.

**Formal analysis:** Sheeza Bano, Noor Khan, Mahroze Fatima, Murat Arslan, Sadia Nazir, Muhammad Asghar, Ayesha Khizar, Alex H. L. Wan.

**Funding acquisition:** Noor Khan, Alex H. L. Wan.

**Investigation:** Sheeza Bano, Mahroze Fatima, Murat Arslan, Alex H. L. Wan.

**Methodology:** Sheeza Bano, Noor Khan, Mahroze Fatima, Murat Arslan, Simon John Davies, Alex H. L. Wan.

**Project administration:** Sheeza Bano, Noor Khan, Mahroze Fatima, Anjum Khalique, Sadia Nazir.

**Resources:** Noor Khan, Murat Arslan, Alex H. L. Wan.

**Software:** Murat Arslan, Alex H. L. Wan.

**Supervision:** Noor Khan, Mahroze Fatima, Anjum Khalique, Murat Arslan, Alex H. L. Wan.

**Validation:** Sheeza Bano, Noor Khan, Mahroze Fatima, Murat Arslan, Simon John Davies, Alex H. L. Wan.

**Visualization:** Sheeza Bano, Noor Khan, Murat Arslan, Sadia Nazir, Muhammad Asghar, Ayesha Khizar, Alex H. L. Wan.

**Writing – original draft:** Sheeza Bano, Noor Khan, Mahroze Fatima, Murat Arslan, Simon John Davies, Alex H. L. Wan.

**Writing – review & editing:** Sheeza Bano, Noor Khan, Mahroze Fatima, Anjum Khalique, Murat Arslan, Sadia Nazir, Muhammad Asghar, Ayesha Khizar, Simon John Davies, Alex H. L. Wan.

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
