## [Decision Letter · Decision Letter 0]

6 Sep 2023

PONE-D-23-19595Enhancing farmed striped catfish (Pangasianodon hypophthalmus) robustness using dietary β-glucanPLOS ONE

Dear Dr. Wan,

Thank you for submitting your manuscript to PLOS ONE. After careful consideration, we feel that it has merit but does not fully meet PLOS ONE’s publication criteria as it currently stands. Therefore, we invite you to submit a revised version of the manuscript that addresses the points raised during the review process.

We look forward to receiving your revised manuscript.

Kind regards,

Ishtiyaq Ahmad, Ph.D

Academic Editor

PLOS ONE

“The authors would like to thank the Punjab Agricultural Research Board (PARB) for providing financial assistance for sample analysis, feed, and experimental fish under the Project: Interactive Effects of Manipulated Artificial Feeds on Growth and Breeding Potential of Channa spp. Special thanks to the Faculty of Fisheries, Ataturk University, Erzurum, Turkey, for the technical support in the fatty acids analysis. Open access funding  for the publication of this research manuscript was provided by IReL open access scholarship.”

“This study was funded by Punjab Agricultural Research Board (PARB), Pakistan- Project: Interactive Effects of Manipulated Artificial Feeds on Growth and Breeding Potential of Channa spp. The funders had no role in study design, data collection and analysis, decision to publish, or preparation of the manuscript. Open access was funded by IREL.”

Reviewers' comments:

Reviewer's Responses to Questions

**Comments to the Author**

1. Is the manuscript technically sound, and do the data support the conclusions?

Reviewer #1: No

Reviewer #2: Partly

2. Has the statistical analysis been performed appropriately and rigorously? 

Reviewer #1: No

Reviewer #2: No

3. Have the authors made all data underlying the findings in their manuscript fully available?

Reviewer #1: Yes

Reviewer #2: No

4. Is the manuscript presented in an intelligible fashion and written in standard English?

Reviewer #1: No

Reviewer #2: Yes

5. Review Comments to the Author

Reviewer #1: Data indicates that higher dietary glucan levels enhance growth performance. However, the authors limited the beta glucan inclusion to 1.5%, necessitating justification for this choice.

Please condense and enhance the conclusion section for clarity.

Kindly ensure the addition of references on Pangasiodon in the reference section

Reviewer #2: I have made a preliminary review of your manuscript and see that the manuscript requires fundamental revisions before it may be reconsidered by reviewers. That is because one-way ANOVA and mean separation is not appropriate analysis for quantitative independent variable such as graded levels of MOS. A more appropriate analysis is to analyze all of the responses using polynomial orthogonal contrasts so you can indicate whether there are significant linear or quadratic responses. The data presented do not support a conclusion or advice for an optimum dose. I strongly recommend reanalysis using either broken line analysis or a linear plateau approach and thus complete revision of the paper for all analyzed parameters. Subsequently, the abstract, results, and discussion sections should be revised and edited to reveal the dose dependent effects of MOS on striped catfish performances.

Furthermore the title of this article does not attract readers’ interest. I found that the author has studied many indicators in this research, but not all indicators must appear in the title.

6. PLOS authors have the option to publish the peer review history of their article (what does this mean?). If published, this will include your full peer review and any attached files.

Reviewer #1: **Yes: **Amit Ranjan

Reviewer #2: No

---

## [Author Response · Author response to Decision Letter 0]

31 Dec 2023

Review Comments to the Author

Reviewer #1: Data indicates that higher dietary glucan levels enhance growth performance. However, the authors limited the beta glucan inclusion to 1.5%, necessitating justification for this choice.

Authors’ response: The following justification was added at line 97

‘The design of the feed trial was to establish the optimum level of β-glucan. Previous study by [16], had shown that 1 g kg-1 was able to elicit a positive response and similarly, other studies had found the optimum level was 1 g kg-1 as compared to other lowest or highest levels [17-21]. Therefore, the following graded levels were tested 0 (G0), 0.5 (G0.5), 1.0 (G1.0), and 1.5 (G1.5) g kg-1 using β-glucan derived from yeast (Food Chem, Huzhou, China).’ However, when tested in the present study, this was not the case, and the upper threshold was not established. The merit of attaining this level is discussed in the discussion section in line 423-432.

Reviewer #1: ‘Please condense and enhance the conclusion section for clarity.

Reviewer’s response: As requested, we have revised the conclusion to the following: 

Line 426-434

‘β-glucan is a potent and effective functional feed ingredient often used in aquafeeds as an elicitor for increasing immunological response. The present study assessed whether dietary β-glucan can offer additional benefits in farmed striped catfish. Using a dietary inclusion level of 1.5 g kg-1 β-glucan in the feed produced the highest beneficial response in growth performance, feed utilisation, body protein accretion, and indicators of antioxidative function. More importantly, the greatest advantage of using dietary β-glucan in striped catfish is its ability to reduce the impact of husbandry stress indicators. However, the tested levels of β-glucan in the present study did not establish the maximal efficacy threshold. Therefore, there is an opportunity for further optimisation in striped catfish.’ 

.

Kindly ensure the addition of references on Pangasiodon in the reference section 

Authors’ response: The name Pangasius is often used for this species as the genus. However, the correct genus name is Pangasiodon and there are papers already cited [14, 16 and 17] under the Pangasiodon species already. In addition, we noticed that Pangasius was used in certain sections of the text instead of the current name- Pangasiodon. We have since made these changes, and are highlighted in the track changes. 

Reviewer #2: I have made a preliminary review of your manuscript and see that the manuscript requires fundamental revisions before it may be reconsidered by reviewers. That is because one-way ANOVA and mean separation is not appropriate analysis for quantitative independent variable such as graded levels of MOS. A more appropriate analysis is to analyze all of the responses using polynomial orthogonal contrasts so you can indicate whether there are significant linear or quadratic responses. The data presented do not support a conclusion or advice for an optimum dose. I strongly recommend reanalysis using either broken line analysis or a linear plateau approach and thus complete revision of the paper for all analyzed parameters. Subsequently, the abstract, results, and discussion sections should be revised and edited to reveal the dose dependent effects of MOS on striped catfish performances.

Authors’ response: We wish to clarify to the reviewer and editor that the test ingredient used in the present study was beta glucan, not MOS- mannooligosaccharide (a prebiotic feed additive). The recommendation of using polynomial orthogonal contrasts is warranted in many dose-response series studies, however, the current study hasn’t established an upper threshold (maximal efficacy). As such, we are unsure whether this would add to the resolution/robustness of the statistical analysis through the use of polynomial orthogonal contrast. Furthermore, this same issue also applies to linear regression analysis. We have added an additional sentence in the discussion line 422-432, stating this. 

‘Furthermore the title of this article does not attract readers’ interest. I found that the author has studied many indicators in this research, but not all indicators must appear in the title.’

Authors’ response: We are unsure what the reviewer is referring to in the manuscript title- ‘studied many indicators in this research, but not all indicators must appear in the title’. Our original submitted title was ‘Enhancing farmed striped catfish (Pangasianodon hypophthalmus) robustness using dietary β-glucan’, which does not list the measured indicators. We believe the proposed title is overarching to reflect the comprehensive nature of the paper.

---

## [Decision Letter · Decision Letter 1]

19 Jan 2024

PONE-D-23-19595R1Enhancing farmed striped catfish (Pangasianodon hypophthalmus) robustness through dietary β-glucanPLOS ONE

Dear Dr. Wan,

Thank you for submitting your manuscript to PLOS ONE. After careful consideration, we feel that it has merit but does not fully meet PLOS ONE’s publication criteria as it currently stands. Therefore, we invite you to submit a revised version of the manuscript that addresses the points raised during the review process.

We look forward to receiving your revised manuscript.

Kind regards,

Ishtiyaq Ahmad, Ph.D

Academic Editor

PLOS ONE

Journal Requirements:

Reviewers' comments:

Reviewer's Responses to Questions

**Comments to the Author**

1. If the authors have adequately addressed your comments raised in a previous round of review and you feel that this manuscript is now acceptable for publication, you may indicate that here to bypass the “Comments to the Author” section, enter your conflict of interest statement in the “Confidential to Editor” section, and submit your "Accept" recommendation.

Reviewer #2: (No Response)

2. Is the manuscript technically sound, and do the data support the conclusions?

Reviewer #2: Yes

3. Has the statistical analysis been performed appropriately and rigorously? 

Reviewer #2: Yes

4. Have the authors made all data underlying the findings in their manuscript fully available?

Reviewer #2: Yes

5. Is the manuscript presented in an intelligible fashion and written in standard English?

Reviewer #2: Yes

6. Review Comments to the Author

Reviewer #2: The title of this article does not attract readers’ interest. I found that the author has studied many indicators in this research, but not all indicators must appear in the title.

7. PLOS authors have the option to publish the peer review history of their article (what does this mean?). If published, this will include your full peer review and any attached files.

Reviewer #2: No

---

## [Author Response · Author response to Decision Letter 1]

24 Jan 2024

In response to point 6 of the Reviewer's response, I believe Reviewer 2 has mistaken the current proposed manuscript with another one. The Reviewer refers to the manuscript title as having excess performance bioindicators but the proposed title doesn't list any indicators, instead it is a concise and informative title. Therefore, I feel that the title doesn't warrant any changes. However, if Reviewer 2 elaborates further or proposes a more suitable title, I would accommodate the suggestion.

---

## [Editor Report · Decision Letter 2]

25 Jan 2024

Enhancing farmed striped catfish (Pangasianodon hypophthalmus) robustness through dietary β-glucan

PONE-D-23-19595R2

Dear Dr. Alex,

We’re pleased to inform you that your manuscript has been judged scientifically suitable for publication and will be formally accepted for publication once it meets all outstanding technical requirements.

Kind regards,

Ishtiyaq Ahmad, Ph.D

Academic Editor

PLOS ONE

---

## [Editor Report · Acceptance letter]

5 Mar 2024

PONE-D-23-19595R2 

PLOS ONE

Dear Dr. Wan, 

I'm pleased to inform you that your manuscript has been deemed suitable for publication in PLOS ONE. Congratulations! Your manuscript is now being handed over to our production team.

Kind regards, 

on behalf of

Dr. Ishtiyaq Ahmad 

Academic Editor

PLOS ONE